# Coloring Learning for Heterophilic Graph Representation

**Miaomiao Huang**[1,2], **Yuhai Zhao**[1,2,*], **Zhengkui Wang**[3], **Fenglong Ma**[4],
**Yejiang Wang**[1,2], **Meixia Wang**[1,2], **Xingwei Wang**[1]

[1] School of Computer Science and Engineering, Northeastern University, China
[2] Key Laboratory of Intelligent Computing in Medical Image
of Ministry of Education, Northeastern University, China
[3] InfoComm Technology Cluster, Singapore Institute of Technology, Singapore
[4] College of Information Sciences and Technology, Pennsylvania State University, United States
```
{huangmiaomiao, wangyejiang, meixiawang}@stumail.neu.edu.cn,
         {zhaoyuhai, wangxw}@mail.neu.edu.cn,
    zhengkui.wang@singaporetech.edu.sg, fenglong@psu.edu
```

## Abstract

Graph self-supervised learning aims to learn the intrinsic graph representations from unlabeled data, with broad applicability in areas such as computing networks. Although graph contrastive learning (GCL) has achieved remarkable progress by generating perturbed views via data augmentation and optimizing sample similarity, it performs poorly in heterophilic graph scenarios (where connected nodes are likely to belong to different classes or exhibit dissimilar features). In heterophilic graphs, existing methods typically rely on random or carefully designed augmentation strategies (e.g., edge dropping) for contrastive views. However, such graph structures exhibit intricate edge relationships, where topological perturbations may completely alter the semantics of neighborhoods. Moreover, most methods focus solely on local contrastive signals while neglecting global structural constraints. To address these limitations, inspired by graph coloring, we propose a novel **Co**loring learning for heterophilic graph **Rep**resentation framework, CoRep, which: 1) Pioneers a coloring classifier to generate coloring labels, explicitly minimizing the discrepancy between homophilic nodes while maximizing that of heterophilic nodes. A global positive sample set is constructed using multi-hop same-color nodes to capture global semantic consistency. 2) Introduces a learnable edge evaluator to guide the coloring learning dynamically and utilizes the edges' triplet relations to enhance its robustness. 3) Leverages Gumbel-Softmax to differentially discretize color distributions, suppressing noise via a redundancy constraint and enhancing intra-class compactness. Experimental results on 14 benchmark datasets demonstrate that CoRep significantly outperforms current state-of-the-art methods.

## 1   Introduction

Self-supervised graph representation learning aims to extract effective low-dimensional representations from graphs without label supervision, which has been widely applied in various fields, such as bioinformatics, social networks, and computing networks [48, 33]. In recent years, graph contrastive learning (GCL) has been identified as one of the most promising self-supervised graph learning methods [8, 32]. GCL primarily consists of two core components: data augmentation and contrastive loss. The former employs various augmentation techniques to create perturbed views for an anchor

---

*Corresponding author.

39th Conference on Neural Information Processing Systems (NeurIPS 2025).

graph, while the latter maximizes the similarity between two views of the same anchor (i.e., positive pairs) and minimizes the similarity between two views from different anchors (i.e., negative pairs).

Although effective for graphs with strong homophily (where adjacent nodes commonly share similar labels or features) [50], these methods fall short when applied to heterophilic graphs. In heterophilic graphs, connected nodes may belong to different classes or exhibit dissimilar attributes [3, 30]. This property is prevalent in many real-world graph structures. For example, in molecular networks, protein structures are typically formed by covalently bonded amino acids of various types [47]. Similarly, in online transaction networks, fraudsters tend to establish links with legitimate users [36]. Recent studies have increasingly focused on graph contrastive learning under heterophily. Existing methods typically explore two main directions: structural decoupling and adaptive augmentation strategies. Structure decoupling-based approaches [27, 48] explicitly separate homophilic and heterophilic structures within the graph based on node or topological similarity, and apply random perturbations (e.g., edge dropping) to each type of structure to construct contrastive views. In contrast, adaptive augmentation-based approaches [44, 7, 6] aim to generate augmented views through carefully designed strategies or learnable generators to better preserve heterophilic connection patterns.

However, the approaches above still suffer from two inherent limitations. First, they heavily rely on random or carefully designed augmentation strategies. However, for heterophilic graphs with complex adjacency structures, such augmentation is challenging and fragile, as it may drastically alter the semantics of the neighborhood. For example, in an online transaction network (as illustrated in Figure 1(a)), fraudsters (brown nodes) tend to establish connections with legitimate users (blue nodes). Applying topological perturbations to such a highly heterophilic structure (as shown in Figure 1(b)) may weaken the sparse yet crucial intra-group connections among fraudsters (brown-brown connections), thereby concealing their collaborative behavior. Moreover, it may mistakenly introduce connections among users (blue-blue connections), disrupting the clear fraud patterns. Such alterations can significantly destroy the semantics of neighborhoods and hinder the model from identifying the underlying behaviors of fraudsters. Second, most approaches concentrate solely on local signals to enforce node-level alignment, which neglects global structural constraints. Consequently, models may overemphasize the consistency of the same node across views while overlooking important relationships among semantically related nodes, thereby sacrificing overall intra-class cohesiveness and inter-class discrimination.

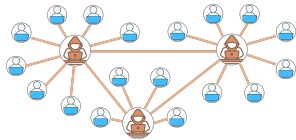

(a) An online transaction network with brown fraudsters and blue users.

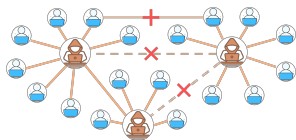

(b) Topological perturbations disrupt the key connections.

Figure 1: Illustrations of an online transaction network.

To address the challenges, we propose a novel **Co**loring learning for heterophilic graph **Rep**resentation framework (CoRep) that aims to assign colors to nodes within the graph such that the colors of adjacent nodes reflect their type differences. Specifically, unlike GCL approaches that depend on data augmentation, CoRep proposes to employ a coloring classifier to generate similar coloring labels to homophilic nodes to explicitly encourage their representations to be close, while assigning different coloring labels to heterophilic nodes to push their representations apart. To dynamically guide the coloring learning, we introduce a learnable edge evaluator that integrates feature and structural information to identify the property of node pairs, while utilizing the edges' triplet relationship to enhance its robustness. Furthermore, we use the Gumbel-Softmax technique for differentiable discretization of color distributions, combined with a sparsity-inducing redundancy constraint to suppress noise and enhance intra-class compactness. To capture global structural consistency, we construct the positive sample set using multi-hop same-color neighbors, thereby ensuring that distant yet semantically related nodes are aligned. Our main contributions can be summarized as follows:

- We propose a CoRep framework for heterophilic graph representation learning through the generated coloring labels, which effectively captures both local and global structures without relying on delicate augmentation strategies.
- CoRep leverages a learnable edge evaluator and a global positive sample set to capture homophily and heterophily more precisely. Moreover, it utilizes a Gumbel-Softmax trick for differentiable discretization, along with a sparsity constraint to enhance intra-class compactness.
- We conduct extensive experiments on 14 benchmark datasets, ranging from relatively citation networks to Wikipedia networks. Experimental results demonstrate that CoRep consistently surpasses state-of-the-art homophilic and heterophilic graph learning methods.

## 2 Related Work

**Graph Neural Networks with Heterophily.** Heterophilic graphs have been widely observed in various scenarios, such as dating networks, online transaction networks, and molecular networks [16, 36, 47, 45]. To better model heterophily structures, recent approaches have proposed a series of Graph Neural Network (GNN) models [25] based on different aggregation mechanisms, including adaptive message propagation [9, 49, 43, 26], high-frequency signal exploration [4, 13, 23], ego and neighbor separation [57, 58], and latent neighbor discovery [20, 54, 10]. Despite their success, these methods often rely on external supervision signals. However, high-quality labels are often scarce in real-world settings. Unlike these methods, this paper focuses on self-supervised learning for heterophilic graphs, aiming to generate discriminative node representations without label supervision.

**Self-supervised Learning on Graphs.** Graph self-supervised learning (SSL) has been a promising paradigm for learning representations without labels [46, 12, 11]. Early studies often utilize random walks or graph reconstruction [51, 1] for graph embedding, but they may lose topological information. GCL methods [18, 59, 53] have attracted considerable attention, aiming to maximize similarity between positive pairs. However, such methods are built upon a strong homophily assumption and perform poorly under heterophily. Until recently, people started to explore SSL on heterophilic graphs. These methods generate perturbed views by leveraging random [27, 48, 56] or carefully designed augmentation strategies [44, 7, 6], then align the augmented positive pairs. However, they heavily rely on effective augmentations, where perturbations to the topology may significantly alter the semantic relationships of neighbors. Differentially, we perform SSL by assigning distinct colors to different types of nodes, fully preserving the structural properties of heterophilic graphs. See Appendix A.1 for more details.

**Graph Coloring.** Graph coloring problem (GCP) is one of the most classical problems in graph theory [15, 24], and has received much attention in many real-world applications, e.g., air traffic flow management [2], register allocation [55], and job scheduling [5]. Its objective is to find a way to assign colors (i.e., coloring labels) to the nodes of a graph such that no two adjacent nodes share the same color while using as few colors as possible. Schuetz et al. [37] proposes to treat graph coloring as a multiclass node classification task and utilize a Potts model for unsupervised learning. In our work, the notion of coloring is used as a heuristic inspiration rather than solving the classical GCP directly. Instead of enforcing distinct colors for adjacent nodes, we extend the coloring concept to heterophilic graph learning: nodes are encouraged to share the same or different colors according to their homophilic or heterophilic relations. This relaxation, combined with a learnable edge evaluator, allows us to capture both affinity and disparity between nodes, thereby facilitating effective representation learning on heterophilic graphs. See Appendix A.2 for more details.

## 3 Methodology

In this section, we elaborate on our **Co**loring learning for heterophilic graph **Rep**resentation (CoRep). The core design of CoRep is illustrated in Figure 2, which comprises three key components: *edge evaluation*, *edge-aware coloring matching learning*, and *multi-hop neighborhood contrastive learning*. In *edge evaluation* module, we introduce a learnable edge evaluator that evaluates the properties of node pairs to dynamically guide the coloring learning (Section 3.3). In *edge-aware coloring matching learning* module, we learn a coloring classifier to generate coloring labels that explicitly encourage similarity between homophilic nodes and dissimilarity between heterophilic nodes (Section 3.4). In *edge-aware coloring matching learning* module, we construct the positive sample set using multi-hop same-color neighbors to capture global structural consistency (Section 3.5).

### 3.1 Notations and Problem

Let $\mathcal{G} = (\mathcal{V}, \mathcal{E})$ denote an undirected graph, where $\mathcal{V} = \{v_1, \ldots, v_n\}$ represents the set of $n$ nodes and $\mathcal{E} \subseteq \mathcal{V} \times \mathcal{V}$ represents the set of edges. The adjacency matrix and the node feature matrix are denoted as $\mathsf{A} \in \{0,1\}^{n \times n}$ and $\mathsf{X} \in \mathbb{R}^{n \times d}$, respectively, where $\mathsf{A}_{ij} = 1$ if $(v_i, v_j) \in \mathcal{E}$, $\mathsf{x}_i \in \mathbb{R}^d$ is the raw feature of node $v_i \in \mathcal{V}$, and $d$ is the input feature dimension. The normalized graph Laplacian matrix is defined as $\mathsf{L} = \mathsf{I}_n - \mathsf{D}^{-1/2}\mathsf{A}\mathsf{D}^{-1/2}$, where $\mathsf{D} \in \mathbb{R}^{n \times n}$ is a diagonal degree matrix with $\mathsf{D}_{i,i} = \sum_j \mathsf{A}_{i,j}$ and $\mathsf{I}_n$ denotes the identity matrix. Let $[\![n]\!] = \{1, \ldots, n\} \subset \mathbb{N}$. For node $v_i$, $\mathscr{N}(v_i) = \{v_j \in \mathcal{V} | (v_i, v_j) \in \mathcal{E}\}$ is its neighbors, and $\mathscr{D}(v_i) := |\mathscr{N}(v_i)|$ is its degree. We define $|\cdot|$

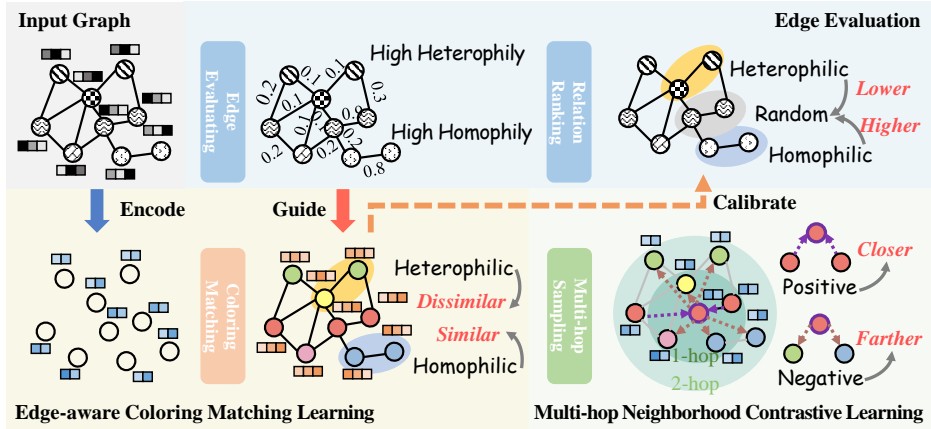

Figure 2: A framework of CoRep.

as the number of elements, $[\cdot \parallel \cdot]$ represents the concatenation operation. In this work, we focus on solving the node-level self-supervised graph representation learning problem. Given $\mathcal{G} = (\mathcal{V}, \mathcal{E})$, we aim to learn an encoder $f_\theta : \mathcal{G} \to \mathbb{R}^{n \times d^\dagger}$ $(d^\dagger \ll d)$ in an unsupervised manner to map the nodes in $\mathcal{G}$ into the $d^\dagger$-dimensional representations, where $\theta$ denotes the parameter of the encoder. These representations preserve graph structures, which can be utilized for downstream tasks, like node classification.

## 3.2 Coloring Matching Learning

In general, obtaining distinguishable node representations in heterophilic graphs without externally supervised signals is a challenging problem. The primary reason lies in the intricate and interwoven connections within heterophilic graphs. Previous GCL methods [27, 48, 44, 7, 6] typically rely on either random or carefully designed augmentation strategies, which tend to cause a complete alteration of neighborhood semantics. For instance, as illustrated in Figure 1, disconnecting links between fraudsters can obscure fraudulent group behaviors. A natural idea is to leverage the inherent properties of heterophilic graphs for representation learning. Fortunately, graph coloring, which assigns different colors (i.e., coloring labels) to adjacent nodes, offers an effective way for our purpose. Inspired by this, we propose a coloring matching learning scheme as a preliminary exploration of coloring learning for heterophilic graph representation. We first encode the structural information of the graph, and then leverage coloring matching to prompt the model to learn distinct coloring labels for adjacent nodes of different types, thereby better adapting to the heterophily structure.

**Structural Encoding.** We first employ an adaptive GNN [4] as the graph encoder $f_\theta : \mathcal{G} \to \mathbb{R}^{n \times d^\dagger}$ to extract node representations. $f_\theta$ utilizes an attention mechanism to adaptively capture both low- and high-frequency signals in $\mathcal{G}$, enabling the effective aggregation from different neighbors. Considering the importance of positional information in recognizing heterophily, we introduce the positional encoding [14] into the attention mechanism to enhance global structural awareness. Each node $v_i$ receives a $d^\sharp$-dimensional position encoding $\mathsf{p}_i$ through $d^\sharp$ steps random walk-based diffusion:

$$\mathsf{p}_i = \left[ \mathsf{T}_{i,i}, \mathsf{T}^2_{i,i}, \ldots, \mathsf{T}^{d^\sharp}_{i,i} \right] \in \mathbb{R}^{d^\sharp} \tag{1}$$

where $\mathsf{T} = \mathsf{A}\mathsf{D}^{-1}$ represents the diffusion transition matrix. See Appendix F.4 for additional details regarding positional encoding. The attention mechanism is defined as:

$$\check{\mathsf{a}}^{(l)}_{i,j} = \tanh\left( \vec{\mathsf{g}}^{(l)\top} \left[ \psi_1\left( \tilde{\mathsf{h}}^{(l)}_i \right) \parallel \psi_1\left( \tilde{\mathsf{h}}^{(l)}_j \right) \right] \right) \tag{2}$$

where $\tilde{\mathsf{h}}^{(l)}_i = \mathsf{h}^{(l)}_i + \psi_0(\mathsf{p}_i)$, $\mathsf{h}^{(l)}_i \in \mathbb{R}^{d^\dagger}$ is the node representation at the $l$-th iteration, $\psi_0 : \mathbb{R}^{d^\sharp} \to \mathbb{R}^{d^\dagger}$ and $\psi_1 : \mathbb{R}^{d^\dagger} \to \mathbb{R}^{d^\dagger}$ are mapping layers, $\vec{\mathsf{g}}^{(l)}$ denotes a weight, and $\tanh(\cdot)$ is an activation function. The node representation $\mathsf{h}^{(l+1)}_i$ at the $(l+1)$-th iteration is updated in a message-passing manner as:

$$\mathsf{h}^{(l+1)}_i = \xi \mathsf{h}^{(0)}_i + \sum_{v_j \in \mathcal{N}(v_i)} \frac{\check{\mathsf{a}}^{(l)}_{i,j}}{\sqrt{\mathscr{D}(v_i)\mathscr{D}(v_j)}} \mathsf{h}^{(l)}_j \tag{3}$$

where $h_i^{(0)} = \psi_2(x_i)$ denotes a transformed node feature by applying a nonlinear mapping layer $\psi_2 : \mathbb{R}^d \to \mathbb{R}^{d^\dagger}$, $\mathscr{N}(v_i)$ is the neighbors of node $v_i$, $\mathscr{D}(v_i)$ is the degree of $v_i$, and $\xi$ is a scaling hyperparameter. The output of the last layer $h_i^{(L)}$ is denoted as $\vec{h}_i$, where $L$ is the number of layers.

**Coloring Matching.** After obtaining the node representations, our objective is to ensure they are well-suited to the heterophily property. Excitingly, we observe that the objective of the graph coloring problem is highly aligned with our learning goal, providing a novel insight for addressing structural heterophily. Graph coloring aims to assign different colors to adjacent nodes while minimizing the number of colors. Let $\chi_{\mathcal{G}} \in \mathbb{N}$ denote the number of available colors, the coloring function $\zeta_{col} : \mathcal{V} \to [\![\chi_{\mathcal{G}}]\!]$ assigns a color to node $v_i$, where $[\![\chi_{\mathcal{G}}]\!] = \{1, \ldots, \chi_{\mathcal{G}}\}$. To evaluate the validity of a coloring scheme, the conflict function $\varsigma^{\mathcal{G}} : \mathcal{V} \times \mathcal{V} \to \{0, 1\}$ is defined as follows:

$$\varsigma^{\mathcal{G}}(v_i, v_j) = \begin{cases} 1, & \text{if } \zeta_{col}(v_i) = \zeta_{col}(v_j) \text{ and } (v_i, v_j) \in \mathcal{E} \\ 0, & \text{otherwise} \end{cases} \tag{4}$$

The objective of graph coloring is to learn a coloring function $\zeta_{col}$ that minimizes the total conflicts $\mathbb{E}[\varsigma^{\mathcal{G}}(v_i, v_j)]$ while reducing the number of used colors $\chi_{\mathcal{G}}$. However, this solution fails to address two fundamental issues: 1) *The inherent diversity of relational patterns.* Heterophilic graphs commonly encompass both heterophilic and homophilic connections, yet this solution cannot distinguish these connection patterns and semantic relationships effectively. 2) *The intractability of direct optimization.* The graph coloring task is inherently a computationally complex combinatorial optimization task, making its direct application to heterophilic graph representation learning difficult to solve in practice.

### 3.3 Edge Evaluation

To address the challenge posed by diverse relational patterns, this section introduces an edge evaluation module. Its core idea lies in the accurate identification of homophilic and heterophilic edges, thereby providing dynamic guidance for the coloring learning process. To achieve this, we introduce a learnable edge evaluator $f_\varepsilon : \mathcal{G} \to \mathbb{R}^{n \times n}$ that integrates feature and structural information to estimate the homophily probability of node pairs [27], where $\varepsilon$ denotes the parameter of the evaluator. Specifically, in heterophilic graphs, relying solely on raw features is insufficient to distinguish node relationships, while positional encodings provide complementary global structural information that enhances node distinguishability. Thus, they are fed into two nonlinear feature mapping layers $\phi_1 : \mathbb{R}^{d+d^\sharp} \to \mathbb{R}^{d^\circ}$ and $\phi_2 : \mathbb{R}^{2d^\circ} \to \mathbb{R}$ to estimate the homophily probability $\omega_{i,j}$ for $(v_i, v_j)$:

$$\begin{aligned} x_i' &= \phi_1([x_i \parallel p_i]), x_j' = \phi_1([x_j \parallel p_j]), \\ \omega_{i,j} &= \left(\phi_2([x_i' \parallel x_j']) + \phi_2([x_j' \parallel x_i'])\right)/2 \end{aligned} \tag{5}$$

where $x_i' \in \mathbb{R}^{d^\circ}$ denotes a transformed feature, where $d^\circ$ is mapping dimension. To more accurately represent edge properties, we aim to sample from $\omega_{i,j}$ to obtain a discriminative result. However, this process introduces a non-differentiability issue. To address this, we adopt the Gumbel-Max reparameterization trick [28, 19] to provide a smooth approximation of the sampling process:

$$\hat{\omega}_{i,j} = \text{Sigmoid}\left((\omega_{i,j} + \log \varpi - \log(1 - \varpi))/\tau_m\right) \tag{6}$$

where $\hat{\omega}_{i,j}$ is the homophily score on $(v_i, v_j)$. A higher value implies a higher homophily between $v_i$ and $v_j$, while a lower value indicates higher heterophily. $\varpi \sim \text{Uniform}(0, 1)$ denotes the sampled Gumbel random variate, $\text{Sigmoid}(\cdot)$ is the activation function, and $\tau_m$ denotes the temperature hyperparameter. As $\tau_m$ approaches 0, samples from the Gumbel-Max distribution become binary.

### 3.4 Edge-aware Coloring Matching Learning

The previous section establishes the foundation for our method, CoRep, by identifying the types of node pairs in the graph. In the following, we will assign similar coloring labels to homophilic node pairs to encourage their representations to be close, while assigning different coloring labels to heterophilic node pairs to push them apart. However, as the second challenge discussed in Section 3.2, the combinatorial optimization problem in coloring matching is difficult to solve directly. Motivated by [37], we transform the optimization problem into a penalty-based loss function, and propose a coloring matching loss and a coloring redundancy constraint to guide the model in generating reasonable coloring labels, thus adapting to the complex structure of heterophilic graphs. Furthermore,

based on the generated coloring labels, a triplet relation ranking loss is proposed to calibrate the edge evaluator, enabling it to more accurately capture the relational properties of node pairs.

**Coloring Matching Loss.** We first propose to employ a coloring classifier $f_\varphi : \mathbb{R}^{d^\dagger} \to \mathbb{R}^{\chi_\mathcal{G}}$ that maps each node to a latent label space, yielding soft coloring labels $\pi_i = f_\varphi(\vec{h}_i)$, where $\varphi$ denotes the parameter of the classifier. To quantify conflicts within the graph, we redefine the conflict function $\varsigma^\mathcal{G}$ as a similarity metric function $\Delta : \mathbb{R}^{\chi_\mathcal{G}} \times \mathbb{R}^{\chi_\mathcal{G}} \to [0,1]$, which measures the similarity of pairs of nodes in the label space. Based on the homophily score $\hat{\omega}_{i,j}$, we encourage adjacent homophilic nodes to share similar coloring labels, while heterophilic nodes are assigned dissimilar ones to reduce overall conflicts in the graph. Accordingly, we propose a coloring matching loss as:

$$\mathcal{L}_m = \frac{1}{n} \sum_{v_i \in \mathcal{V}} \frac{1}{|\mathscr{N}(v_i)|} \sum_{v_j \in \mathscr{N}(v_i)} (1 - \hat{\omega}_{i,j}) \cdot \Delta(\pi_i, \pi_j) \tag{7}$$

where $\Delta(\cdot, \cdot)$ denotes a similarity metric function, e.g., cosine similarity. This design in the Equation (7) enables CoRep to directly capture the semantic relations between node pairs based on intrinsic graph structure, leading to more discriminative node representations.

**Coloring Redundancy Constraint.** In practice, the coloring classifier may assign excessive or semantically uninformative color classes when fitting heterophilic structures, leading to redundant colors that introduce noise and reduce intra-class compactness. To mitigate this issue, we leverage the Gumbel-Softmax trick [19] to approximate discrete color sampling in a differentiable manner, allowing for more diverse color assignments, and propose a sparsity-inducing coloring redundancy constraint that encourages each node to be confidently assigned to a more relevant color to suppres diffuse and noisy color distributions:

$$\mathcal{L}_d = \sum_{j \in \llbracket \chi_\mathcal{G} \rrbracket} \Phi_{max} \left( \{\mathscr{C}_{ij}^{col}\}_{i \in \llbracket n \rrbracket} \right), \ \mathscr{C}_{ij}^{col} = \frac{\exp\left( (\log(\pi_{ij}) + \varrho_{ij})/\tau_o \right)}{\sum\limits_{j \in \llbracket \chi_\mathcal{G} \rrbracket} \exp\left( (\log(\pi_{ij}) + \varrho_{ij})/\tau_o \right)} \tag{8}$$

where $\Phi_{max} : \mathbb{R}^n \to \mathbb{R}$ is a column-wise max pooling operator, $\varrho_{ij}$ denotes a sampled Gumbel random variate and $\tau_o$ is a temperature parameter. In our work, we set a small $\tau_o$ to encourage $\mathscr{C}_{ij}^{col}$ to approach a one-hot vector. $\mathscr{C}_{ij}^{col}$ provides an effective way to explore diverse color assignments to avoid deterministic selection that may lead to suboptimal local minima.

**Triplet Relation Ranking Loss.** The coloring learning process described above relies on the reliability of edge relations. However, accurately evaluating these connections without supervision signals is inherently difficult. To dynamically calibrate the edge evaluation, we design a triplet relation ranking loss that controls the deviation between the homophily and heterophily degrees of node pairs and the similarity of their assigned coloring labels, ensuring that the output of the evaluator accurately reflects their relationships. Specifically, we randomly sample a node pair $(v_p, v_q)$ from the input graph and aim for all node pairs satisfy the following relationship: $\forall (v_i, v_j) \in \mathscr{E}_{i,j}^+ \succcurlyeq (v_p, v_q) \succcurlyeq \forall (v_i, v_j) \in \mathscr{E}_{i,j}^-$, where $\succcurlyeq$ represents an ordering relation such that $\iota_0 \succcurlyeq \iota_1$ means $\iota_0$ is ranked before $\iota_1$, $\mathscr{E}_{i,j}^+$ and $\mathscr{E}_{i,j}^-$ represent the sets of homophilic and heterophilic edges:

$$\mathscr{E}_{i,j}^+ = \left\{ (v_i, v_j) \mid (v_i, v_j) \sim \Pi^{homo}(\hat{\omega}_{i,j}) \right\}, \ \mathscr{E}_{i,j}^- = \left\{ (v_i, v_j) \mid (v_i, v_j) \sim \Pi^{hete}(1 - \hat{\omega}_{i,j}) \right\} \tag{9}$$

where $\Pi^{homo}(\hat{\omega}_{i,j})$ and $\Pi^{hete}(1 - \hat{\omega}_{i,j})$ denote the probability distribution based on the scores $\hat{w}_{i,j}$ and $1 - \hat{w}_{i,j}$, respectively. $\Pi^{homo}(\hat{\omega}_{i,j})$ is used to sample node pairs with higher homophily, while $\Pi^{hete}(1 - \hat{\omega}_{i,j})$ is used to sample node pairs with higher heterophily. By leveraging the above triplet relation ranking to ensure that the similarity of node pairs with homophily is ranked higher than randomly sampled node pairs, while the similarity of node pairs with heterophily is ranked lower than randomly sampled node pairs, we can dynamically calibrate the joint optimization of edge evaluation and representation learning. To enforce this ordering, we propose a triplet relation ranking loss to penalize the ranking errors of node pairs:

$$\mathcal{L}_r = -\sum_{e_{i,j} \in \mathcal{E}} (1 - \hat{\omega}_{i,j}) \log\left( \sigma\left( \Delta_{p,q}^{rnd} - \Delta_{i,j}^{mat} \right) \right) + \hat{\omega}_{i,j} \log\left( 1 - \sigma\left( \Delta_{p,q}^{rnd} - \Delta_{i,j}^{mat} \right) \right) \tag{10}$$

where $\Delta_{i,j}^{mat} := \Delta(\pi_i, \pi_j)$ and $\Delta_{p,q}^{rnd} := \Delta(\pi_p, \pi_q)$ denote the semantic similarity between $(v_i, v_j)$ and between $(v_p, v_q)$, respectively.

## 3.5 Multi-hop Neighborhood Contrastive Learning

Due to the inherent heterophily of the graph structure, semantically similar nodes are often distributed in non-adjacent topological regions. Relying only on locally connected neighbors is insufficient to capture the global semantic consistency. To address this, we design a robust multi-hop neighborhood contrastive learning module that compels our model to preserve the semantic consistency of long-range neighbor nodes. This module identifies multi-hop neighbors with the same semantics through positive sample selection and aligns their representations using a contrastive loss.

**Positive Sample Selection.** To identify distant yet semantically related nodes, we propose to construct the global positive sample set using multi-hop neighbor nodes that share the same colors. Specifically, we leverage $\mathscr{C}_i^{col}$ and $\mathscr{C}_j^{col}$ to determine whether two nodes $v_i$ and $v_j$ are semantically related. Based on this, the semantically related $\varkappa$-hop neighbor nodes are selected as the positive sample set:

$$\mathcal{P}_{\mathcal{G}}(v_i) = \left\{ v_j \mid v_j \in \mathcal{N}^{(\varkappa)}(v_i), \ \mathscr{C}_j^{col} = \mathscr{C}_i^{col} \right\} \tag{11}$$

where $\mathcal{N}^{(\varkappa)}(v_i)$ denotes the neighbor nodes of node $v_i$ within $\varkappa$ hops, which is defined as follows: $\mathcal{N}^{(\varkappa)}(v_i) = \mathcal{N}(v_i)$ if $\varkappa = 1$; $\mathcal{N}^{(\varkappa)}(v_i) = \mathcal{N}^{(\varkappa-1)}(v_i) \cup \left( \bigcup_{v_k \in \mathcal{N}^{(\varkappa-1)}(v_i)} \mathcal{N}(v_k) \right)$ if $\varkappa > 1$.

**Multi-hop Neighborhood Contrastive Loss.** Given the positive sample set above, we further design a multi-hop neighborhood contrastive loss, which aims to bring distant yet semantically related multi-hop neighbor nodes closer, while pushing apart other nodes. Specifically, we utilize a projector $f_\upsilon : \mathbb{R}^{d^\dagger} \to \mathbb{R}^{d^\natural}$ to map each node representation into a $d^\natural$-dimensional latent representation $\mathsf{z}_i = f_\upsilon(\vec{\mathsf{h}}_i)$ for a fair comparison, where $\upsilon$ denotes the parameter of the projector. Inspired by the InfoNCE contrastive loss [59], the multi-hop neighborhood contrastive loss is defined as:

$$\mathcal{L}_c = -\frac{1}{n} \sum_{v_i \in \mathcal{V}} \frac{1}{|\mathcal{P}_{\mathcal{G}}(v_i)|} \sum_{v_j \in \mathcal{P}_{\mathcal{G}}(v_i)} \log \frac{\exp\left(\mathrm{sim}(\mathsf{z}_i, \mathsf{z}_j)/\tau_c\right)}{\sum\limits_{v_k \in \mathcal{V} \backslash \mathcal{P}_{\mathcal{G}}(v_i)} \exp\left(\mathrm{sim}(\mathsf{z}_i, \mathsf{z}_k)/\tau_c\right)} \tag{12}$$

where $\mathrm{sim}(\cdot, \cdot)$ denotes consine similarity, and $\tau_c$ is a temperature hyperparameter.

## 3.6 Overall Loss

As the coloring label distribution approaches uniformity, node representations become less distinguishable. We introduce an entropy regularization term $\mathcal{L}_e = \sum_{i \in [\![n]\!]} \pi_i^T \log \pi_i$ to encourage the generation of more definitive coloring labels. Hence, the overall loss is then formulated as:

$$\mathcal{L} = \mathcal{L}_m + \alpha \mathcal{L}_d + \beta \mathcal{L}_r + \gamma \mathcal{L}_c + \eta \mathcal{L}_e \tag{13}$$

where $\alpha$, $\beta$, $\gamma$, and $\eta$ are trade-off hyperparameters. We present the overall algorithm in Appendix B.

## 3.7 Complexity Analysis

In this section, we analyze the time complexity of CoRep. Let $|\mathcal{V}|$ and $|\mathcal{E}|$ be the number of nodes and edges. For the computation of positional encodings and multi-hop neighborhoods, their complexities are $\mathcal{O}(|\mathcal{V}||\mathcal{E}|d^\natural)$ and $\mathcal{O}(|\mathcal{V}||\mathcal{E}|\varkappa)$. Note that these two steps are computed at once. For the edge evaluator and graph encoder, their costs are $\mathcal{O}(|\mathcal{V}|d^\circ(d + d^\natural) + |\mathcal{E}|d^\circ)$ and $\mathcal{O}(Ld^\dagger(|\mathcal{V}| + |\mathcal{E}|))$. CoRep contains three core losses: $\mathcal{L}_m$, $\mathcal{L}_r$, and $\mathcal{L}_c$, and their complexities are $\mathcal{O}(\chi_{\mathcal{G}}|\mathcal{E}|)$, $\mathcal{O}(\chi_{\mathcal{G}}|\mathcal{E}|)$, and $\mathcal{O}(|\mathcal{V}|b\kappa d^\natural)$, where $\kappa$ denotes the average number of positive samples ($\kappa \ll |\mathcal{V}|$), and $b$ is the batch size of the loss. Detailed complexity analysis can be found in Appendix C.

# 4 Experiments

This section empirically evaluates the proposed CoRep method on 14 benchmark datasets and analyzes its behavior on graphs to gain further insights. More results can be found in the Appendix F.

## 4.1 Experimental Setup

**Datesets.** To assess the quality of the learned representations, we employ transductive node classification as the downstream task. Our experiments are conducted on 14 widely used benchmark

Table 1: Results in terms of classification accuracies (in percent $\pm$ standard deviation) on homophilic benchmarks. The best and second-best performance under each dataset are marked with **boldface** and underline, respectively. OOM indicates Out-Of-Memory.

| Methods | Cora | CiteSeer | PubMed | Wiki-CS | Computers | Photo | CS | Physics |
|---|---|---|---|---|---|---|---|---|
| GCN | 81.50±1.30 | 70.30±0.28 | 78.80±2.90 | 76.89±0.37 | 86.34±0.48 | 92.35±0.25 | 93.10±0.17 | 95.54±0.19 |
| GAT | 82.80±1.30 | 71.50±0.49 | 78.50±0.27 | 77.42±0.19 | 87.06±0.35 | 92.64±0.42 | 92.41±0.27 | 95.45±0.17 |
| MLP | 56.11±0.34 | 56.91±0.42 | 71.35±0.05 | 72.02±0.21 | 73.88±0.10 | 78.54±0.05 | 90.42±0.08 | 93.54±0.05 |
| H2GCN | 80.23±0.20 | 69.97±0.66 | 78.79±0.30 | 79.73±0.13 | 84.32±0.52 | 91.86±0.27 | 91.18±0.58 | 93.56±0.48 |
| FAGCN | 77.80±0.66 | 69.81±0.80 | 76.74±0.66 | 74.34±0.53 | 83.51±1.04 | 92.72±0.22 | 93.81±0.24 | 96.16±0.15 |
| PC-Conv | 82.47±0.56 | 69.92±1.33 | 79.57±1.23 | 79.94±0.52 | 87.89±0.26 | **93.89±0.14** | 94.24±0.12 | 95.99±0.14 |
| DeepWalk | 69.47±0.55 | 58.82±0.61 | 69.87±1.25 | 74.35±0.06 | 85.68±0.06 | 89.44±0.11 | 84.61±0.22 | 91.77±0.15 |
| node2vec | 71.24±0.89 | 47.64±0.77 | 66.47±1.00 | 71.79±0.05 | 84.39±0.08 | 89.67±0.12 | 85.08±0.03 | 91.19±0.04 |
| GAE | 71.07±0.39 | 65.22±0.43 | 71.73±0.92 | 70.15±0.01 | 85.27±0.19 | 91.62±0.13 | 90.01±0.71 | 94.92±0.07 |
| VGAE | 79.81±0.87 | 66.75±0.37 | 77.16±0.31 | 75.63±0.19 | 86.37±0.21 | 92.20±0.11 | 92.11±0.09 | 94.52±0.00 |
| DGI | 82.29±0.56 | 71.49±0.14 | 77.43±0.84 | 75.73±0.13 | 84.09±0.39 | 91.49±0.25 | 91.95±0.40 | 94.57±0.38 |
| GMI | 82.51±1.47 | 71.56±0.56 | 79.83±0.90 | 75.06±0.13 | 81.76±0.52 | 90.72±0.33 | OOM | OOM |
| MVGRL | 83.03±0.27 | 72.75±0.46 | 79.63±0.38 | 77.97±0.18 | 87.09±0.27 | 92.01±0.13 | 91.97±0.19 | 95.53±0.10 |
| GRACE | 80.08±0.53 | 71.41±0.38 | 80.15±0.34 | 79.16±0.36 | 87.21±0.44 | 92.65±0.32 | 92.78±0.23 | 95.39±0.32 |
| GCA | 80.39±0.42 | 71.21±0.24 | 80.37±0.75 | 79.35±0.12 | 87.84±0.27 | 92.78±0.17 | 93.32±0.12 | 95.87±0.15 |
| BGRL | 81.08±0.17 | 71.59±0.42 | 79.97±0.36 | 78.74±0.22 | 88.92±0.33 | 93.24±0.29 | 93.26±0.36 | 95.76±0.38 |
| HGRL | 80.66±0.43 | 68.56±1.10 | 80.35±0.58 | 76.68±0.17 | 84.30±0.47 | 93.53±0.22 | 93.99±0.15 | OOM |
| GREET | 83.32±0.49 | 72.20±1.01 | 80.50±0.66 | 79.87±0.49 | 87.55±0.37 | 92.99±0.38 | 94.68±0.21 | 95.91±0.14 |
| HeteGCL | 81.55±0.65 | 70.63±1.16 | 82.50±0.57 | 79.12±0.25 | 85.76±0.21 | 93.82±0.32 | **94.79±0.06** | OOM |
| CoRep | **85.04±0.34** | **73.67±0.40** | **83.50±0.47** | **82.20±0.51** | **89.17±3.81** | 93.84±1.89 | 94.39±0.31 | **96.21±0.11** |

datasets, consisting of 8 homophilic graph datasets (i.e., Cora, CiteSeer, PubMed, Wiki-CS, Amazon Computers, Amazon Photo, CoAuthor CS, and CoAuthor Physics) [38, 29, 39] and 6 heterophilic graph datasets (i.e., Chameleon, Squirrel, Actor, Cornell, Texas, and Wisconsin) [31]. The statistics of all datasets are summarized in Appendix D.

**Baselines.** We compare CoRep with 5 groups of baseline methods, including 1) supervised/semi-supervised learning methods (i.e. GCN [22], GAT [42], and MLP), 2) supervised learning methods specially designed for heterophilic graphs (i.e. H2GCN [57], FAGCN [4], and PC-Conv [23]), 3) conventional unsupervised graph representation learning methods (i.e., DeepWalk [35], node2vec [17], GAE, and VGAE [21]), 4) contrastive self-supervised learning methods (i.e., DGI [41], GMI [34], MVGRL [18], GRACE [59], GCA [60], and BGRL [40]), and 5) contrastive self-supervised learning methods designed for heterophilic graphs (i.e., HGRL [6], GREET [27], and HeteGCL [44]).

**Evaluation Protocol.** For CoRep and all unsupervised baselines, we follow the standard linear evaluation protocol of previous state-of-the-art graph self-supervised learning approaches at the node classification task [50, 6], where a linear classifier is trained on top of the frozen representation, and test accuracy is used as a proxy for representation quality. For datasets, we adopt the standard dataset splits used in previous studies, i.e., public splits [52, 22, 31] or commonly used splits [60, 27].

**Experimental Details.** All methods were implemented in PyTorch with the Adam Optimizer. We run 10 times of experiments and report the average test accuracy with standard deviation. For fair comparison, the parameters of all baselines are tuned according to the parameter ranges reported by the authors. Specific hyperparameter settings and more implementation details are in Appendix E.

## 4.2 Performance Comparison

Table 1 and Table 2 display the node classification results for 8 homophilic datasets and 6 heterophilic datasets, respectively. Comparing the results in Tables 1 and 2, we have the following major observations. First, we find that our CoRep outperforms all baseline methods in 10 out of 14 benchmarks and achieves the second and third-best performance on the remaining 4 benchmarks. For example, CoRep achieves accuracies of 85.04% and 82.20% on the Cora and Wiki-CS datasets, respectively, which is a relative improvement of over 1.72% and 2.26% compared to the best baselines. For heterophilic graphs, CoRep achieves a relative improvement of over 5.71% and 2.97% on the Squirrel and Texas datasets compared to the best baselines. The superior performance indicates that coloring learning on heterophilic graphs can produce expressive and generalizable representations. Moreover, we also observe that CoRep significantly outperforms conventional and contrastive unsupervised graph learning methods, surpasses heterophily-oriented unsupervised learning methods in 85.71% of cases, and outperforms supervised learning methods under heterophily

Table 2: Results in terms of classification accuracies (in percent ± standard deviation) on heterophilic benchmarks. The best and second-best performance under each dataset are marked with **boldface** and underline, respectively. OOM indicates Out-Of-Memory.

| Methods | Chameleon | Squirrel | Actor | Cornell | Texas | Wisconsin |
|---|---|---|---|---|---|---|
| GCN | 59.63±2.32 | 36.28±1.52 | 30.83±0.77 | 57.03±3.30 | 60.00±4.80 | 56.47±6.55 |
| GAT | 56.38±2.19 | 32.09±3.27 | 28.06±1.48 | 59.46±3.63 | 61.62±3.78 | 54.71±6.87 |
| MLP | 46.91±2.15 | 29.28±1.33 | 35.66±0.94 | 81.08±7.93 | 81.62±5.51 | 84.31±3.40 |
| H2GCN | 59.39±1.98 | 37.90±2.02 | 35.86±1.03 | 82.16±4.80 | 84.86±6.77 | 86.67±4.69 |
| FAGCN | 63.44±2.05 | 41.17±1.94 | 36.81±0.26 | 81.35±5.05 | 84.32±6.02 | 83.33±2.01 |
| PC-Conv | 53.20±1.60 | 35.79±0.62 | 36.07±0.61 | 78.65±2.70 | 85.68±2.97 | **88.63±2.94** |
| DeepWalk | 47.74±2.05 | 32.93±1.58 | 22.78±0.64 | 39.18±5.57 | 46.49±6.49 | 33.53±4.92 |
| node2vec | 41.93±3.29 | 22.84±0.72 | 28.28±1.27 | 42.94±7.46 | 41.92±7.76 | 37.45±7.09 |
| GAE | 33.84±2.77 | 28.03±1.61 | 28.03±1.18 | 58.85±3.21 | 58.64±4.53 | 52.55±3.80 |
| VGAE | 35.22±2.71 | 29.48±1.48 | 26.99±1.56 | 59.19±4.09 | 59.20±4.26 | 56.67±5.51 |
| DGI | 39.95±1.75 | 31.80±0.77 | 29.82±0.69 | 63.35±4.61 | 60.59±7.56 | 55.41±5.96 |
| GMI | 46.97±3.43 | 30.11±1.92 | 27.82±0.90 | 54.76±5.06 | 50.49±2.21 | 45.98±2.76 |
| MVGRL | 51.07±2.68 | 35.47±1.29 | 30.02±0.70 | 64.30±5.43 | 62.38±5.61 | 62.37±4.32 |
| GRACE | 48.05±1.81 | 31.33±1.22 | 29.01±0.78 | 54.86±6.95 | 57.57±5.68 | 50.00±5.83 |
| GCA | 49.80±1.81 | 35.50±0.91 | 29.65±1.47 | 55.41±4.56 | 59.46±6.16 | 50.78±4.06 |
| BGRL | 47.46±2.74 | 32.64±0.78 | 29.86±0.75 | 57.30±5.51 | 59.19±5.85 | 52.35±4.12 |
| HGRL | 48.29±1.64 | 35.79±0.89 | 36.97±0.98 | 79.46±4.45 | 82.16±6.00 | 86.28±3.58 |
| GREET | 63.09±2.18 | 40.86±1.93 | 35.75±1.08 | 73.78±3.64 | 85.41±3.67 | 84.12±4.76 |
| HeteGCL | 48.77±1.55 | 34.27±1.58 | **37.59±1.22** | 81.32±6.26 | 82.37±5.83 | 80.39±5.23 |
| CoRep | **65.64±1.39** | **46.88±1.56** | 37.32±1.13 | **82.70±4.55** | **88.65±3.97** | 86.86±3.17 |

in 85.71% of cases as well. This result suggests that goal consistency between coloring learning and heterophilic graph learning can facilitate the model's effective adaptation to heterophily structures.

## 4.3 Ablation Study

To examine the contribution of key designs in CoRep, we consider the following ablations. (**A1**) We remove the edge-aware coloring matching learning (w/o Col. Mat.), where the hard assignment $argmax_{j \in [\![\chi_{\mathcal{G}}]\!]} \pi_{i,j}$ of predicted coloring labels is directly used to guide positive sample selection in multi-hop neighborhood contrastive learning. (**A2**) We remove the learnable edge evaluator (w/o Ed. Eva.), where the coloring matching loss is computed directly from the coloring labels without using edge evaluation. (**A3**) We remove the Gumbel-Softmax technique (w/o Gum. Soft.), where the hard assignment $argmax_{j \in [\![\chi_{\mathcal{G}}]\!]} \pi_{i,j}$ replaces the node's color $\mathscr{C}_i^{col}$ in Equation (11) to identify the positive sample set. (**A4**) We remove the coloring redundancy constraint (w/o $\mathcal{L}_d$) by setting $\alpha = 0$. (**A5**) We remove the triplet relation ranking loss (w/o $\mathcal{L}_r$) by setting $\beta = 0$. (**A6**) We remove the multi-hop neighborhood contrastive loss (w/o $\mathcal{L}_c$) by setting $\gamma = 0$. We show the ablation study results in Table 3 (More results can be found in the Appendix F.1). From Table 3, we can see that A1 has a significant impact on the model's performance, highlighting the importance of edge-aware coloring matching learning. The introduction of A2 and A3 further improves its performance, indicating the effectiveness of edge evaluation and the Gumbel-Softmax. The performance degradation observed in A4, A5, and A6 highlights the critical role of the loss terms $\mathcal{L}_d$, $\mathcal{L}_r$, and $\mathcal{L}_c$ in CoRep, as they are essential for maintaining intra-class compactness, edge discriminability, and global structural consistency, respectively. We also observe that the combination of losses $\mathcal{L}_d$, $\mathcal{L}_r$, and $\mathcal{L}_c$ yields better results compared to using each loss individually. The complete model (last row) achieves the best performance, demonstrating that the different components of the proposed CoRep framework are complementary and work synergistically.

Table 3: Ablation studies results (mean classification accuracy) on the Cornell and Texas datasets.

| Ablation | Cornell | Texas |
|---|---|---|
| **A1** w/o Col. Mat. | 70.00 | 77.57 |
| **A2** w/o Ed. Eva. | 76.49 | 85.95 |
| **A3** w/o Gum. Soft. | 77.57 | 82.16 |
| **A4** w/o $\mathcal{L}_d$ | 79.73 | 85.95 |
| **A5** w/o $\mathcal{L}_r$ | 81.62 | 86.49 |
| **A6** w/o $\mathcal{L}_c$ | 75.14 | 84.32 |
| **A4+A5** | 78.92 | 85.14 |
| **A4+A6** | 73.78 | 83.78 |
| **A5+A6** | 74.59 | 78.65 |
| **CoRep** | **82.70** | **88.65** |

## 4.4 Parameter Analysis

In this section, we perform a detailed sensitivity analysis on the number of colors and neighborhood hops. Additional hyperparameter experiments can be found in Appendix F.2.

**Effect of the Number of Colors** $\chi_{\mathcal{G}}$**.** We vary $\chi_{\mathcal{G}}$ from 5 to 20 with a 5-unit interval to examine its impact on the model. The classification accuracies under different $\chi_{\mathcal{G}}$ choices are shown in Figure 3(a). We observe that the best results across CiteSeer, Cornell, and Texas datasets often occur when $\chi_{\mathcal{G}}$ is relatively large. This phenomenon suggests that appropriately increasing the number of colors helps the model uncover underlying semantic information. Meanwhile, the coloring redundancy constraint ensures intra-class compactness to avoid introducing irrelevant colors.

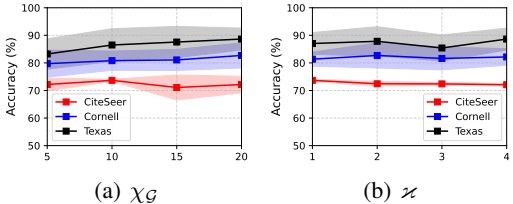

Figure 3: Parameter sensitivity of $\chi_{\mathcal{G}}$ and $\varkappa$.

**Effect of the Neighborhood Hops** $\varkappa$**.** We then vary neighborhood hops $\varkappa$ to investigate the model's sensitivity to the neighborhood scope. As shown in Figure 3(b), we observe that superior performance is obtained using only one-hop neighbors on the CiteSeer dataset, suggesting that local structures are sufficient to reflect class consistency in highly homophilic graphs. In contrast, on the Cornell and Texas datasets, relying on multi-hop neighbors leads to better performance, indicating that in highly heterophilic graphs, incorporating broader structural information helps preserve features of distant yet semantically related nodes, thereby maintaining global semantic consistency.

## 5    Conclusion

In this paper, inspired by graph coloring, we proposed a coloring learning for heterophilic graph representation (CoRep). Unlike prevailing GCL approaches that rely heavily on carefully designed augmentations, our method focuses on assigning distinct colors to different types of nodes. Specifically, we: 1) Pioneered a coloring classifier to generate similar/dissimilar coloring labels to homophilic/heterophilic nodes to encourage them to be closer/farther; Constructed a global positive sample set using multi-hop same-color neighbors to capture global structural consistency. 2) Introduced a learnable edge evaluator to guide the coloring learning dynamically and utilized edges' triplet relationship to enhance its robustness. 3) Leveraged the Gumbel-Softmax and redundancy constraint to enhance intra-class compactness. Extensive experiments on 14 benchmark datasets showed the effectiveness of CoRep. The limitations and broader impacts of CoRep are discussed in Appendix H.

## Acknowledgments

We thank the anonymous reviewers for their constructive suggestions. This project was in part supported by the following projects: the National Natural Science Foundation of China (No.62432003).

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
