# OpenReview forum: "Coloring Learning for Heterophilic Graph Representation"
_NeurIPS.cc/2025/Conference — NeurIPS 2025 poster_

### Official Review · Reviewer_e6Mg · 2025-06-23

**Clarity:** 4
**Significance:** 3
**Originality:** 3
**Rating:** 4
**Confidence:** 4

**Summary:**

This paper introduces CoRep, a self-supervised framework for heterophilic graph representation learning. Arguing that traditional augmentation-based GCL methods damage crucial heterophilic structures, CoRep proposes a novel pretext task inspired by graph coloring. Instead of altering the graph, the model learns to assign different latent "colors" to adjacent dissimilar nodes and similar colors to adjacent similar nodes. This is achieved through a framework that includes a learnable edge evaluator to guide the process, an edge-aware coloring matching loss to learn representations, and a multi-hop contrastive module to enforce global consistency between distant, same-colored nodes. Extensive experiments on 14 benchmark datasets demonstrate that CoRep achieves state-of-the-art performance.

**Questions:**

See Weaknesses.

**Ethical Concerns:**

["NO or VERY MINOR ethics concerns only"]

**Final Justification:**

This paper draws inspiration from the graph coloring problem and introduces a self-supervised learning approach designed for heterophilic graphs. The idea is interesting and provides a fresh perspective by addressing some of the difficulties that traditional GCL methods encounter in data augmentation for heterophilic graphs. I think these contributions are meaningful and deserve acceptance.

**Limitations:**

Yes.

**Paper Formatting Concerns:**

None.

**Quality:**

3

**Strengths And Weaknesses:**

Strengths

1. Redefining the classic Graph Coloring Problem (GCP) and applying it to self-supervised heterophilic graph learning is a novel idea. It elegantly circumvents the difficulties faced by traditional GCL methods in performing data augmentation on heterophilic graphs, providing a new perspective for addressing this issue.
2. The design of the CoRep framework is comprehensive, integrating multiple modules that work collaboratively.
3. The authors conduct extensive experiments on 14 datasets, and the results are convincing, supporting the effectiveness of their approach.
4. The paper is clearly written and easy to understand.

Weaknesses

1. The edge evaluator relies solely on the original node features $ x $ and positional encodings $ p $ to predict edge homogeneity. This method appears to depend heavily on the quality of the input features. If the original features $ X $ are noisy or not sufficiently discriminative, how does the edge evaluator avoid failure? Could more direct structural information, such as the number of common neighbors or higher-order neighborhood structures, be incorporated to assist in the assessment?

2. The core argument of the paper is its design for heterophilic graphs. Why does it also perform well on homogeneous graphs? Is it because the edge evaluator can accurately identify that all edges are homogeneous, causing the model to degenerate into a standard homogeneous graph learning method? If so, this framework is not just a "heterophilic graph" solution, but rather a more general one. This universality is worth deeper discussion in the paper to enhance the generality of the work.

3. The authors could more clearly delineate the differences and connections between their notion of "coloring" and the classical GCP in the related work or methods section. Clarifying that it serves as a heuristic idea could help avoid theoretical controversies.

4. The partial order definition after Equation (9) is somewhat obscure. It would be helpful to explain the underlying idea in more intuitive terms: namely, that the color similarity of matched edges should be higher than that of random edges, and the similarity of random edges should, in turn, be higher than that of mismatched edges.

---

> ### Author Rebuttal · Authors · 2025-07-31
>
> > **The edge evaluator relies solely on the original node features $ x $ and positional encodings $ p $ to predict edge homogeneity. This method appears to depend heavily on the quality of the input features. If the original features $ X $ are noisy or not sufficiently discriminative, how does the edge evaluator avoid failure? Could more direct structural information, such as the number of common neighbors or higher-order neighborhood structures, be incorporated to assist in the assessment?**
>
> We appreciate the reviewer’s insightful comment. We agree that the edge evaluator’s discriminative power may degrade when the input node features are noisy or insufficiently informative. However, our model does not rely solely on the raw node features. We also incorporate $d^\sharp$-dimensional positional encodings, which capture structural information up to $d^\sharp$-hop neighborhoods. This integration helps mitigate the weakness of noisy node attributes by introducing richer topological information into the edge evaluation process. In addition, we propose a triplet relation ranking loss that explicitly measures and penalizes the discrepancy between the predicted edge homophily score and node coloring labels. This auxiliary supervision further regularizes the learning of the edge evaluator and enhances its robustness.
>
> To better understand the impact of noisy input, we conduct a robustness analysis by adding Gaussian noise to the inputs of the edge evaluator during training at each epoch. As shown in Table I, although the model performance degrades when 20% or 50% noise is added, the overall drop is relatively small, indicating that the edge evaluator maintains strong robustness under noisy conditions.
>
> We also acknowledge the reviewer’s suggestion to incorporate more explicit structural cues, such as the number of common neighbors or higher-order neighborhood patterns. This is a promising direction, and we plan to explore it in future work to further enhance edge assessment reliability.
>
> Table I. Model performance under Gaussian noise in edge evaluator inputs.
> |Noise Ratio|Cornell|Texas|Chameleon|Photo|CS|
> |-----------|-------|-----|---------|-----|---|
> |20%|82.16±2.16|85.41±4.71|65.57±1.50|93.79±1.83|94.37±0.34|
> |50%|80.54±5.51|84.86±4.39|65.44±1.26|93.77±1.78|94.33±0.33|
> |CoRep|**82.70±4.55**|**88.65±3.97**|**65.64±1.39**|**93.84±1.89**|**94.39±0.31**|
>
> > **The core argument of the paper is its design for heterophilic graphs. Why does it also perform well on homogeneous graphs? Is it because the edge evaluator can accurately identify that all edges are homogeneous, causing the model to degenerate into a standard homogeneous graph learning method? If so, this framework is not just a "heterophilic graph" solution, but rather a more general one. This universality is worth deeper discussion in the paper to enhance the generality of the work.**
>
> We are truly grateful for the reviewers’ insightful understanding of our method and their valuable recognition. While our primary motivation is to address the challenges posed by heterophily, our design is inherently general. Specifically, our approach leverages a learnable edge evaluator that dynamically assesses the relationships between node pairs. In heterophilic graphs, this enables the model to distinguish between different types of neighboring nodes and assign them different scores. In homophilic graphs, the edge evaluator tends to assign similar scores to neighboring nodes with similar labels or features, effectively capturing community structures and preserving label consistency. Moreover, the edge-aware coloring mechanism in our model further supports this adaptability by assigning similar/dissimilar colors to nodes of the same/different types. This flexible design allows our method to act as a general framework capable of handling varying levels of homophily without requiring architectural changes.
>
> To validate this generality, we have conducted thorough experiments on several benchmark homophilic datasets in the paper. As shown in Table 1 (Section 4.2), our method generally performs better than strong baselines. These results confirm that our framework is not limited to heterophilic scenarios but is broadly effective across different graph types.  We will include this discussion in the revisions of the paper.
>
> > **The authors could more clearly delineate the differences and connections between their notion of "coloring" and the classical GCP in the related work or methods section. Clarifying that it serves as a heuristic idea could help avoid theoretical controversies.**
>
> We appreciate the reviewer’s insightful comment. We fully agree that the classical Graph Coloring Problem (GCP) serves as a heuristic inspiration for our work, rather than being directly solved in our method. Specifically, the classical GCP seeks to assign different colors to adjacent nodes while minimizing the total number of colors, which is a well-known NP-hard combinatorial optimization problem and often impractical to solve exactly for large graphs. In contrast, our method extends the concept of coloring to heterophilic graph learning. We do not directly solve the classical GCP; instead, we leverage the coloring idea along with a learnable edge evaluator to assign the same or different colors to node pairs based on their homophilic or heterophilic relations, which achieves representation learning for heterophilic graphs.  This allows the model to effectively capture both affinity and disparity between nodes, that is, the underlying homophilic and heterophilic dependencies and semantics. We will include this discussion in the revisions to better distinguish our approach from the classical GCP.
>
> > **The partial order definition after Equation (9) is somewhat obscure. It would be helpful to explain the underlying idea in more intuitive terms: namely, that the color similarity of matched edges should be higher than that of random edges, and the similarity of random edges should, in turn, be higher than that of mismatched edges.**
>
> Thank you for the insightful comment. We will incorporate this clarification in the revised version of the paper. Specifically, we will explain that the partial order is meant to reflect a similarity ranking: homophilic edges should have higher similarity than random edges, which in turn should be more similar than heterophilic ones. This encourages the model to better distinguish different types of node relations. Thanks again.

---

> > ### Comment · Reviewer_e6Mg · 2025-08-03
> >
> > Thanks for the authors' response. My concerns are well addressed. I encourage the authors to incorporate the discussions from the rebuttal  into the next revision.

---

> > > ### Author Response · Authors · 2025-08-04
> > > **Thank you for your thoughtful review!**
> > >
> > > We sincerely appreciate your valuable feedback and are glad that your concerns have been addressed. Your comments were instrumental in improving the paper. We will incorporate the relevant discussions and clarifications into the final version.

---

### Official Review · Reviewer_1zrg · 2025-07-02

**Clarity:** 3
**Significance:** 3
**Originality:** 4
**Rating:** 5
**Confidence:** 4

**Summary:**

The paper proposes a self-supervised heterophilic graph representation framework based on coloring learning.  Unlike augmentation-based GCL approaches that rely on complex augmentation strategies, this work introduces the coloring classifier and learnable edge evaluator to generate adaptive coloring labels to effectively capture both local and global heterophilic structures.  It achieves the state-of-the-art performance across multiple datasets.

**Questions:**

**Questions**

1.In the parameter analysis experiment, the CiteSeer dataset demonstrates the best performance with one-hop neighbors. It would be valuable if the authors could provide further analysis or insights into why this is the case, particularly in terms of the dataset's characteristics or the model's behavior, which might explain the observed performance trend.

2.The method introduced in this paper performs excellently on both homophilic and heterophilic graphs, demonstrating strong adaptability to various graph structures. However, could the authors discuss the scalability of their model on larger graphs, such as ogbn-arxiv?

3.How does the runtime of your method compare to other baseline approaches? A comparison of computational efficiency across different methods would provide valuable insights into the practicality and feasibility of your method.

**Ethical Concerns:**

["NO or VERY MINOR ethics concerns only"]

**Final Justification:**

I have read all the reviews and response. My concerns have been fully addressed. I think the authors present an interesting perspective on self-supervised graph representation learning, which offers valuable insights to the field. The experimental results are strong and convincing. Therefore, I am in favor of accepting this paper.

**Limitations:**

yes

**Paper Formatting Concerns:**

There are no major formatting issues in this paper. The manuscript adheres to the NeurIPS 2025 Paper Formatting Instructions.

**Quality:**

3

**Strengths And Weaknesses:**

**Strengths**

S1. Novel Integration of coloring learning and heterophilic graph learning. The framework innovatively proposes coloring learning to generate different coloring labels for different node pairs to explore local and global heterophilic structures, offering a meaningful solution for unsupervised heterophilic graph learning.

S2. The paper is well-written and easy to understand. The illustrations and problem descriptions in the paper are well-matched and clearly presented, effectively highlighting the problem the paper aims to address.

S3. The proposed method demonstrates good adaptability, performing consistently across both homophilic and heterophilic graphs. This suggests its robustness to varying graph structures.

S4. Comprehensive experimental validation. The effectiveness of the proposed method is well-supported through extensive comparisons with a wide range of baseline approaches across numerous datasets.

**Weakness**

W1. The explanation of the triplet relation ranking loss lacks clarity. It would be helpful to elaborate on what the partial order among the three elements represents, and how this ordering is reflected in the design of the ranking loss.

W2. Lack of some setting details: Some hyperparameters, such as the temperature parameter $\tau$ and the dimensionality of the position encoding, are not provided in experiments.

---

> ### Author Rebuttal · Authors · 2025-07-31
>
> > **The explanation of the triplet relation ranking loss lacks clarity. It would be helpful to elaborate on what the partial order among the three elements represents, and how this ordering is reflected in the design of the ranking loss.**
>
> Thank you for the suggestion. In the triplet constraint
> $\forall\left(v_i, v_j\right)\in\mathscr{E}_{i, j}^{+}\succcurlyeq\left(v_p, v_q\right)\succcurlyeq\forall\left(v_i, v_j\right)\in\mathscr{E} _{i, j}^{-}$,
>
> the three elements represent node pairs sampled from the sets of homophilic edges, randomly selected edges, and heterophilic edges, respectively. The partial order among them is designed to establish a relative ranking, such that the semantic similarity of homophilic pairs is expected to be higher than that of randomly sampled pairs, which in turn should be higher than that of heterophilic pairs. This ordering is explicitly enforced through the design of the triplet relation ranking loss, which penalizes violations of the expected relative similarity among the three types of relations. Specifically, the loss function penalizes cases where the similarity of a randomly sampled node pair  $(v_p, v_q)$ exceeds that of a homophilic pair $(v_i, v_j)$ (where a high $\widehat
> {\omega}_ {i, j} $ indicates stronger homophily), as well as cases where the similarity of $(v_p, v_q)$ is lower than that of a heterophilic pair (where a low $\widehat{\omega}_{i, j}$ indicates stronger heterophily). In this way, the evaluator is encouraged to better distinguish homophilic and heterophilic relations. We will clarify this ordering relation and its implementation in the revised version.
>
> > **Lack of some setting details: Some hyperparameters, such as the temperature parameter $\tau$ and the dimensionality of the position encoding, are not provided in experiments.**
>
> Thank you for the valuable feedback.  In our experiments, we set the temperature parameters $\tau_m$, $\tau_o$, and $\tau_c$ to 1.0, 1.0, and 0.2, respectively. The dimensionality of the position encoding $d^\sharp$ is set to 16, and the hidden dimension of the edge evaluator $d^\circ$ is set to 128. We will include these details in the revised version of the paper.
>
> > **In the parameter analysis experiment, the CiteSeer dataset demonstrates the best performance with one-hop neighbors. It would be valuable if the authors could provide further analysis or insights into why this is the case, particularly in terms of the dataset's characteristics or the model's behavior, which might explain the observed performance trend.**
>
> Thank you for your valuable suggestion. We analyze this phenomenon from two perspectives: dataset characteristics and model behavior. From the perspective of dataset characteristics, compared to more heterophilic graphs such as Texas and Cornell, the CiteSeer dataset has a relatively simple structure and exhibits strong homophily (with an edge homophily score of 0.736). This high level of homophily indicates that one-hop neighbors already provide sufficient class-consistent information, offering clear and effective semantic supervision for representation learning. From the model behavior perspective, contrastive learning relies on constructing semantically consistent positive pairs from a node's neighbors to enhance the discriminative power of node representations. While incorporating multi-hop neighbors can help capture global semantic relationships, in strongly homophilic graphs like CiteSeer, distant neighbors often span class boundaries and are less semantically relevant. Treating these multi-hop nodes as positive samples may introduce false positives, thereby weakening the effectiveness of the contrastive loss and even hindering the training process. As a result, restricting contrastive neighbors to the one-hop level proves more beneficial for learning high-quality representations on CiteSeer.
>
> > **The method introduced in this paper performs excellently on both homophilic and heterophilic graphs, demonstrating strong adaptability to various graph structures. However, could the authors discuss the scalability of their model on larger graphs, such as ogbn-arxiv?**
>
> We conducted experiments on two large-scale datasets: OGBN-Arxiv (homophilic) and Arxiv-Year (heterophilic). Both datasets are large in scale, containing 169,343 nodes and 1,166,243 edges. Table I presents the node classification results on two large-scale datasets. As shown in Table I, our method achieves the best performance on Arxiv-Year and ranks second on OGBN-Arxiv, closely following BGRL. These results confirm the effectiveness and scalability of our approach on large-scale graphs.
>
> Table I. Results on homophilic and heterophilic large-scale graphs. OOM indicates Out-Of-Memory.
> | Methods   | OGBN-Arxiv   | Arxiv-Year    |
> |-----------|--------------|---------------|
> | node2vec  | 70.07±0.13   | 39.69±0.09    |
> | DGI       | 70.19±0.73   | 40.60±0.21    |
> | MVGRL     | OOM          | OOM           |
> | BGRL      | **71.24±0.35**   | 41.43±0.04    |
> | HGRL      | OOM          | OOM           |
> | GREET     | OOM          | OOM           |
> | CoRep     | 70.59±0.07   | **42.59±0.42** |
>
> > **How does the runtime of your method compare to other baseline approaches? A comparison of computational efficiency across different methods would provide valuable insights into the practicality and feasibility of your method.**
>
> We appreciate the reviewer’s suggestion. The runtime of our method compared to baseline approaches has already been presented in Table II. Experimental results show that our method achieves comparable running speed to BGRL, one of the most efficient graph contrastive learning methods, and even outperforms it on larger datasets such as PubMed, CS, and Physics. This improvement is primarily attributed to our use of sparse computation and mini-batch processing. Furthermore, compared to other baseline methods, our approach demonstrates significant efficiency advantages. For example, on the Physics dataset, our method is approximately 45 times faster than GREET.  We will include these results in the revised version of the paper.
>
> Table II. The average elapsed time per training epoch of the methods (in seconds).
> | Methods   | CiteSeer | PubMed | CS     | Physics | Cornell | Texas  |
> |-----------|----------|--------|--------|---------|---------|--------|
> | MVGRL     | 0.343    | 0.345  | 2.004  | OOM     | 0.034   | 0.034  |
> | BGRL      | **0.033** | 0.182  | 0.202  | 0.568   | **0.022** | **0.022** |
> | HGRL      | 0.058    | 0.836  | 1.770  | OOM     | 0.031   | 0.037  |
> | GREET     | 0.202    | OOM    | OOM    | 21.891  | 0.025   | 0.025  |
> | HeteGCL   | 0.154    | 0.851  | 1.627  | OOM     | 0.061   | 0.061  |
> | CoRep     | 0.047    | **0.156** | **0.164** | **0.491** | **0.022**   | 0.023  |

---

> > ### Comment · Reviewer_1zrg · 2025-08-08
> >
> > Thank you for the detailed response and for the AC’s reminder. I have read all the reviews and response. My concerns have been fully addressed. I think the authors present an interesting perspective on self-supervised graph representation learning, which offers valuable insights to the field. The experimental results are strong and convincing. Therefore, I am in favor of accepting this paper.

---

> > > ### Author Response · Authors · 2025-08-08
> > > **Appreciation for Your Feedback!**
> > >
> > > We are pleased to have addressed the reviewers’ comments and will incorporate the corresponding clarifications into the final version. We sincerely thank the reviewers for their valuable suggestions, which have helped improve the quality of our work.

---

### Official Review · Reviewer_aqBg · 2025-07-03

**Clarity:** 3
**Significance:** 3
**Originality:** 4
**Rating:** 5
**Confidence:** 4

**Summary:**

In this paper, the authors propose a new method “CoRep” for heterophilic graph representation learning. CoRep seeks to capture local and global structures via coloring learning instead of graph contrastive learning. It generates coloring labels to encourage homophilic nodes to be closer and push heterophilic nodes farther apart. A learnable edge evaluator and a global positive sample set are used to capture graph structure more precisely. Extensive experiments on 14 benchmark datasets show the superiority of this method.

**Questions:**

1. The method proposed by the authors is designed for heterophilic graph structures. Could the authors clarify whether the proposed approach is specifically tailored to heterophilic graphs and whether its applicability to homophilic graphs is limited? A discussion on the generalizability of the method across different types of graph structures would strengthen the paper.
2. The authors employ two Gumbel distributions in the proposed method. Could the authors elaborate on whether these two distributions serve distinct roles within the modeling process?
3. The paper introduces redundancy constraints to reduce “redundant colors”, but does not explain what “redundant” means (e.g., in complex structures, assigning more colors may be semantically necessary rather than noise).

**Ethical Concerns:**

["NO or VERY MINOR ethics concerns only"]

**Limitations:**

yes

**Quality:**

4

**Strengths And Weaknesses:**

Strengths:
1. This paper proposes a novel coloring learning framework for heterophilic graph representation learning. This idea is interesting. It provides a different perspective from the mainstream data augmentation paradigm to solve the heterophilic graph problem, directly modeling the core feature of "different neighbor node categories" in heterophilic graphs.
2. This paper is written clearly and well organized. The presentation of examples and illustrations is very clear and easy to understand.
3. The proposed method is intuitive and easy to follow.
4. Extensive experimental studies show the effectiveness of this method, providing support for their conclusions.

Weaknesses:
1. This paper lacks experiments on large-scale datasets to demonstrate the scalability of the proposed method.
2. Some concepts are not clearly illustrated. See the questions below.

---

> ### Author Rebuttal · Authors · 2025-07-31
>
> > **This paper lacks experiments on large-scale datasets to demonstrate the scalability of the proposed method.**
>
> Thank you for your valuable suggestion. We have conducted experiments on two large-scale datasets: the homophilic dataset OGBN-Arxiv and the heterophilic dataset Arxiv-Year. Both datasets are large in scale, containing 169,343 nodes and 1,166,243 edges. The experimental results are presented in Table I. From Table I, we can see that our proposed method ranks second on the OGBN-Arxiv dataset, slightly behind BGRL, and achieves the best performance on the Arxiv-Year dataset. These results demonstrate that our method maintains strong performance and scalability on large-scale graphs with both homophilic and heterophilic structures. We will add these results in the revised version.
>
> Table I. Results on homophilic and heterophilic large-scale graphs. OOM indicates Out-Of-Memory.
> | Methods   | OGBN-Arxiv   | Arxiv-Year    |
> |-----------|--------------|---------------|
> | node2vec  | 70.07±0.13   | 39.69±0.09    |
> | DGI       | 70.19±0.73   | 40.60±0.21    |
> | MVGRL     | OOM          | OOM           |
> | BGRL      | **71.24±0.35**   | 41.43±0.04    |
> | HGRL      | OOM          | OOM           |
> | GREET     | OOM          | OOM           |
> | CoRep     | 70.59±0.07   | **42.59±0.42** |
>
> > **The method proposed by the authors is designed for heterophilic graph structures. Could the authors clarify whether the proposed approach is specifically tailored to heterophilic graphs and whether its applicability to homophilic graphs is limited? A discussion on the generalizability of the method across different types of graph structures would strengthen the paper.**
>
> We appreciate the reviewer’s insightful comment. Our method is not only applicable to heterophilic graphs but also well-suited for homophilic graphs, demonstrating strong generality and adaptability. Specifically, our approach introduces a learnable edge evaluator that adaptively quantifies the relationships between node pairs. In heterophilic graphs, this component enables the model to effectively differentiate between various types of neighboring nodes by assigning distinct scores, thereby capturing diverse relational patterns. In contrast, in homophilic graphs, the evaluator tends to produce similar scores for neighbors sharing similar features or labels, which helps preserve label consistency and reflect underlying community structures. Moreover, the edge-aware coloring mechanism in our model further supports this adaptability by assigning similar/dissimilar colors to nodes of the same/different types. This flexible design allows our method to act as a general framework capable of handling varying levels of homophily without requiring architectural changes.
>
> To demonstrate its generalizability, we have conducted extensive experiments on multiple homophilic graph datasets in the paper. As shown in Table 1 (Section 4.2), our method achieves superior performance compared to baselines, highlighting its effectiveness beyond heterophilic scenarios.
>
> > **The authors employ two Gumbel distributions in the proposed method. Could the authors elaborate on whether these two distributions serve distinct roles within the modeling process?**
>
> Thank you for the valuable feedback. Although both techniques are based on Gumbel distributions, they are applied to different components of the model to address distinct challenges: Gumbel-Max is used for sampling discrete edge-level homophily indicators, while Gumbel-Softmax is employed for approximating node-level color assignments in a differentiable way. Specifically, the Gumbel-Max trick in Equation (6) is employed to sample the homophily score between node pairs in a differentiable manner. The ambiguity of estimated homophily probabilities makes them difficult to use directly for discrimination, while sampling discrete indicators from them is inherently non-differentiable. By leveraging the Gumbel-Max trick, we achieve approximate sampling of explicit homophily scores, which helps the edge evaluator effectively distinguish between homophilic and heterophilic node pairs. In contrast, the Gumbel-Softmax trick in Equation (8) is used to facilitate more effective color assignments. Due to the structural complexity of heterophilic graphs, the coloring classifier may generate noisy or redundant class assignments. The Gumbel-Softmax trick provides a differentiable approximation to discrete color sampling, encouraging confident and sparse color assignments. This mitigates redundancy and enhances intra-class compactness.
>
> > **The paper introduces redundancy constraints to reduce “redundant colors”, but does not explain what “redundant” means (e.g., in complex structures, assigning more colors may be semantically necessary rather than noise).**
>
> We appreciate the reviewer’s insightful comment regarding the definition of “redundant colors.” In our paper, redundant colors refer to excessive or semantically uninformative color classes that emerge during model training as a result of overfitting to heterophilic structures. These assignments often do not contribute meaningfully to the classification objective and instead reduce intra-class compactness. To address this issue, our proposed redundancy constraint does not arbitrarily limit the number of color classes. Instead, it promotes sparse color distributions at the node level to encourage confident and semantically meaningful color assignments. This is achieved via a differentiable Gumbel-Softmax sampling mechanism, which maintains flexibility in exploring diverse coloring patterns, while the sparsity term guides the model away from diffuse and noisy assignments. We will add this discussion to the revised version.

---

> > ### Comment · Reviewer_aqBg · 2025-08-04
> >
> > Thank you for the rebuttal. The authors' responses have addressed my main concerns. I'll keep my positive score.

---

> > > ### Author Response · Authors · 2025-08-04
> > > **Thank you for your feedback!**
> > >
> > > We’re pleased to hear that your concerns have been resolved. Thank you for your thoughtful review, which helped improve our work. We will reflect the clarifications in the final version.

---

### Official Review · Reviewer_avE4 · 2025-07-03

**Clarity:** 2
**Significance:** 2
**Originality:** 2
**Rating:** 4
**Confidence:** 3

**Summary:**

This paper addresses the limitations of graph contrastive learning (GCL) on heterophilic graphs, where existing methods rely on random augmentations and neglect global structural constraints. The proposed CoRep framework introduces a coloring-based approach: 1) a coloring classifier generates labels to minimize homophilic and maximize heterophilic node discrepancies, with multi-hop same-color nodes forming global positive sets; 2) a learnable edge evaluator guides coloring learning using triplet relationships; 3) Gumbel-Softmax and redundancy constraints enhance intra-class compactness. Experiments on public datasets show CoRep outperforms SOTA methods, with ablation studies validating each component's effectiveness.

**Questions:**

Please refer to the limitations part.

**Ethical Concerns:**

["NO or VERY MINOR ethics concerns only"]

**Final Justification:**

The authors have responded to my comments and the previous concerns were addressed. I would like to keep my rating as weak accept.

**Quality:**

2

**Strengths And Weaknesses:**

Overall, the manuscript does not have significant flaws; however, some methodological details and certain experimental aspects need to be further supplemented and clarified, as outlines below.

1. In Section 3.2, regarding the structural encoding, the authors state that "Each node $v_i$ receives a $d^\#$ \-dimensional position encoding $\mathbf{p}_i$ through $d^\#$ steps random walk-based diffusion" (with Eq. 1). The details and role of this position embedding (PE) require further clarification. Specifically, the authors should specify the value of $d^\#$, and clarify whether the PE, once initialized, is treated as a trainable parameter during the entire network training process. Furthermore, it is important to discuss the impact of PE on both homophilic and heterophilic graphs. An ablation study regarding the effectiveness of PE should be provided.

2. The complexity analysis in Section 3.7 is well presented. However, the authors should also compare the time complexity of the proposed method with that of the baselines. Additionally, reporting the actual time and space complexity in practice (such as training time) would make the results more convincing.

3. The datasets selected for the ablation study (Table 3) are relatively small. Conducting ablation experiments on larger datasets would provide a more comprehensive evaluation of the contribution of each component.

4. The authors should further elaborate on the detailed settings of the ablation experiments, as the current description is somewhat too brief. Please provide detailed configurations for each ablation study. Given the overall framework, the removal of certain key components may affect the subsequent structure, and any ambiguous settings need to be clarified. For example, in the A3 setting, after removing Gumbel-Softmax, how is positive sample selection conducted in Multi-hop Neighborhood Contrastive Learning?

5. The visualization results (Figure 5) are very informative. Considering that this work focuses on unsupervised learning on heterophilic graphs, why not also provide visualization results on a heterogeneous graph to intuitively demonstrate the effectiveness of CoRep for heterophilic graphs?

---

> ### Author Rebuttal · Authors · 2025-07-31
>
> >**In Section 3.2, regarding the structural encoding, the authors state that "Each node vi receives a d^# -dimensional position encoding pi through d^# steps random walk-based diffusion" (with Eq. 1). The details and role of this position embedding (PE) require further clarification. Specifically, the authors should specify the value of d^#, and clarify whether the PE, once initialized, is treated as a trainable parameter during the entire network training process. Furthermore, it is important to discuss the impact of PE on both homophilic and heterophilic graphs. An ablation study regarding the effectiveness of PE should be provided.**
>
> We appreciate the reviewer's suggestion. We clarify the details and certain experiments as follows. (1) Following [1,2], we set the dimension of the position embedding (PE) $d^\sharp$ to 16 in all experiments. This choice achieves a balance between information effectiveness and computational efficiency: it is sufficient to encode meaningful structural information of the graph while keeping memory usage and training overhead within a reasonable range. (2) The PE is computed once via the random walk-based diffusion process and remains fixed (non-trainable) during training. This design allows the PE to serve as a stable structural prior independent of the specific task, which helps prevent overfitting and promotes better generalization across datasets. (3) To evaluate the impact of PE, we conducted an ablation study by removing it from the model in Table I. From Table I, the model's performance drops when PE is removed, highlighting the utility of PE in capturing long-range and structurally meaningful dependencies. We also observe that PE contributes more significantly to heterophilic datasets such as Texas and Cornell, where local feature aggregation is less informative and global structural cues from PE help guide the learning process. We will include this clarification and the ablation study in the revised version.
>
> Table I. An ablation study on position embedding (PE).
> |Ablation|Cornell|Texas|Chameleon|Photo|CS|
> |--------|-------|-----|---------|-----|---|
> |w/o PE|81.08±2.70|85.95±5.24|64.78±1.36|93.20±2.36|94.12±0.23|
> |CoRep|**82.70±4.55**|**88.65±3.97**|**65.64±1.39**|**93.84±1.89**|**94.39±0.31**|
>
> [1] Beyond Smoothing: Unsupervised Graph Representation Learning with Edge Heterophily Discriminating. AAAI 2023.
>
> [2] Graph neural networks with learnable structural and positional representations. ICLR 2022.
>
> >**The complexity analysis in Section 3.7 is well presented. However, the authors should also compare the time complexity of the proposed method with that of the baselines. Additionally, reporting the actual time and space complexity in practice (such as training time) would make the results more convincing.**
>
> Thank you for your valuable suggestion. We have added the theoretical time and space complexities of our method and baseline methods during training in Table II for clearer comparison. Additionally, we report the actual training time of these methods in Table III. The results show that our method achieves comparable running speed to BGRL, one of the most efficient graph contrastive learning methods, and even outperforms it on larger datasets such as PubMed, CS, and Physics. This improvement is primarily attributed to sparse computation and mini-batch processing. Furthermore, compared to other baselines, our approach demonstrates significant efficiency advantages. For example, on the Physics dataset, our method is approximately 45 times faster than GREET. We will include these results in the revised version.
>
> Table II. The time and space complexities of the six methods.
> |Methods|Time Complexity|Space Complexity|
> |-------|----------------|----------------|
> |MVGRL|$\mathcal{O}(\|\mathcal{V}\|^2 d^\dagger+{\|\mathcal{V}\|d^\dagger}^2)$|$\mathcal{O}(\|\mathcal{V}\|^2 +\|\mathcal{V}\|d^\dagger)$|
> |BGRL|$\mathcal{O}(\|\mathcal{E}\| d^\dagger+{\|\mathcal{V}\|d^\dagger}^2)$|$\mathcal{O}(\|\mathcal{E}\| +\|\mathcal{V}\|d^\dagger+{\|\mathcal{V}\|d^\dagger}^2)$|
> |HGRL|$\mathcal{O}(\|\mathcal{V}\|^2 d^\dagger+{d^\dagger}^2)$|$\mathcal{O}(\|\mathcal{V}\|^2 +\|\mathcal{V}\|d^\dagger)$|
> |GREET|$\mathcal{O}(\|\mathcal{E}\| d^\dagger+\|\mathcal{V}\|b \kappa d^\natural+\|\mathcal{V}\|dd^\dagger)$|$\mathcal{O}(\|\mathcal{E}\| +\|\mathcal{V}\|d^\natural)$|
> |HeteGCL|$\mathcal{O}(\|\mathcal{V}\|^2 d^\dagger+{d^\dagger}^2)$|$\mathcal{O}(\|\mathcal{V}\|^2 +\|\mathcal{V}\|d^\dagger)$|
> |CoRep|$\mathcal{O}(\|\mathcal{E}\| d^\dagger+\|\mathcal{V}\|b \kappa d^\natural+\|\mathcal{V}\|dd^\dagger)$|$\mathcal{O}(\|\mathcal{E}\| +\|\mathcal{V}\|d^\natural)$|
>
> where $|\mathcal{V}|$ and $|\mathcal{E}|$ denote the number of nodes and edges in the graph. $d$, $d^\dagger$, and $d^\natural$ are the dimensionality of the original node features, final node representations, and projected node representations. $\kappa$ represents the average number of positive samples. $b$ is the batch size of the contrastive loss.
>
> Table III. The average elapsed time per training epoch of the methods (in seconds). OOM indicates Out-Of-Memory.
> |Methods|CiteSeer|PubMed|CS|Physics|Cornell|Texas|
> |-------|--------|------|---|-------|-------|-----|
> |MVGRL|0.343|0.345|2.004|OOM|0.034|0.034|
> |BGRL|**0.033**|0.182|0.202|0.568|**0.022**|**0.022**|
> |HGRL|0.058|0.836|1.770|OOM|0.031|0.037|
> |GREET|0.202|OOM|OOM|21.891|0.025|0.025|
> |HeteGCL|0.154|0.851|1.627|OOM|0.061|0.061|
> |CoRep|0.047|**0.156**|**0.164**|**0.491**|**0.022**|0.023|
>
> >**The datasets selected for the ablation study (Table 3) are relatively small. Conducting ablation experiments on larger datasets would provide a more comprehensive evaluation of the contribution of each component.**
>
> Thank you for the helpful suggestion. To further validate the effectiveness of each component, we conduct ablation studies on larger datasets. As shown in Table IV, each module positively impacts overall performance. Specifically, we observe that removing A1 leads to a performance drop across all three datasets. The results on the Chameleon and Photo datasets are particularly unstable, indicating A1’s importance in the model. Similarly, drops caused by removing A2 and A3 highlight the key roles of the edge evaluator and the Gumbel-Softmax trick. The performance degradation in A4, A5, and A6 highlights the critical role of the loss terms $\mathcal{L}_d$, $\mathcal{L}_r$, and $\mathcal{L}_c$ in CoRep, as they are essential for maintaining intra-class compactness, edge discriminability, and global structural consistency, respectively. We also observe that the combination of losses yields better results compared to using each loss individually. The complete model (last row) achieves the best performance, demonstrating the complementary and synergistic effects of all components.
>
> Table IV. Ablation experiments on larger datasets.
> |Ablation|Chameleon|Photo|CS|
> |--------|---------|-----|---|
> |A1 w/o Col.Mat.|62.78±7.16|91.32±2.36|93.18±0.35|
> |A2 w/o Ed.Eva.|64.52±1.73|88.71±9.53|94.19±0.12|
> |A3 w/o Gum.Soft.|65.18±1.21|92.24±3.28|93.27±2.44|
> |A4 w/o $\mathcal{L}_d$|65.37±1.70|93.74±1.83|94.21±0.28|
> |A5 w/o $\mathcal{L}_r$|65.31±2.10|93.79±1.82|94.18±0.22|
> |A6 w/o $\mathcal{L}_c$|51.64±7.14|89.62±3.50|89.84±4.25|
> |A4+A5|64.77±1.38|93.20±1.92|93.63±0.40|
> |A4+A6|51.36±8.32|88.92±3.19|88.56±3.91|
> |A5+A6|50.96±7.21|88.90±2.84|88.00±4.40|
> |CoRep|**65.64±1.39**|**93.84±1.89**|**94.39±0.31**|
>
> >**The authors should further elaborate on the detailed settings of the ablation experiments, as the current description is somewhat too brief. Please provide detailed configurations for each ablation study. Given the overall framework, the removal of certain key components may affect the subsequent structure, and any ambiguous settings need to be clarified. For example, in the A3 setting, after removing Gumbel-Softmax, how is positive sample selection conducted in Multi-hop Neighborhood Contrastive Learning?**
>
> We thank the reviewer for this valuable observation.  We clarify the ablation settings as follows:
> (A1) Removing the edge-aware coloring matching module (w/o Col. Mat.). Instead, the hard assignment $argmax_{j\in⟦\chi_{\mathcal{G}}⟧}\pi_{i,j}$ of predicted coloring labels is directly used to guide the positive sample selection in multi-hop neighborhood contrastive learning. (A2) Removing the learnable edge evaluator (w/o Ed.Eva.). In this case, the coloring matching loss is computed directly based on the coloring labels, without relying on the edge evaluation. (A3) Removing the Gumbel-Softmax technique (w/o Gum.Soft.). We use the hard assignment of coloring labels, i.e., $argmax_{j\in⟦\chi_\mathcal{G}⟧}\pi_{i,j}$, to replace the node’s color $\mathscr{C}_i^{col}$ in Equation (11) to identify positive sample set in the contrastive learning. This setting evaluates whether the Gumbel-Softmax technique is essential for selecting positive samples. (A4) Removing the coloring redundancy constraint (w/o $\mathcal{L}_d$) by setting $\alpha=0$. (A5) Removing the triplet relation ranking loss (w/o $\mathcal{L}_r$) by setting $\beta=0$.  (A6) Removing the multi-hop neighborhood contrastive loss (w/o $\mathcal{L}_c$) by setting $\gamma=0$, to examine the role of global semantic consistency. We will include these detailed settings in the revised version.
>
> >**The visualization results (Figure 5) are very informative. Considering that this work focuses on unsupervised learning on heterophilic graphs, why not also provide visualization results on a heterogeneous graph to intuitively demonstrate the effectiveness of CoRep for heterophilic graphs?**
>
> Thank you for your helpful suggestion. We fully agree that visualizing results on a heterophilic graph could further demonstrate the effectiveness of CoRep for heterophily settings. Due to the figure limitations of this conference submission, we were unable to include additional visualizations. We will add such visualization results in the revised version. Thank you again.

---

### Decision · Program_Chairs · 2025-09-17

**Decision:**

Accept (poster)

**Comment:**

The paper introduces a new approach to self-supervised graph learning tailored to heterophilic graphs. The proposed framework follows a novel formulation based on graph coloring, introducing a classifier that assigns similar coloring labels to homophilic nodes. Beyond the originality of this formulation, the methodology is supported by a thorough experimental evaluation on both homophilic and heterophilic benchmark graphs. Overall, the paper makes a solid and well-executed contribution to the field. I encourage the authors to carefully consider the comments of the reviewers to enhance the proposed methodology and its presentation.